# Smart-watch-programmed green-light-operated percutaneous control of therapeutic transgenes

Maysam Mansouri [1], Marie-Didiée Hussherr[1], Tobias Strittmatter [1], Peter Buchmann[1], Shuai Xue[1], Gieri Camenisch[1] & Martin Fussenegger [1,2 ✉]

Wearable smart electronic devices, such as smart watches, are generally equipped with green-light-emitting diodes, which are used for photoplethysmography to monitor a panoply of physical health parameters. Here, we present a traceless, green-light-operated, smart-watch-controlled mammalian gene switch (Glow Control), composed of an engineered membrane-tethered green-light-sensitive cobalamin-binding domain of *Thermus thermophilus* (TtCBD) CarH protein in combination with a synthetic cytosolic TtCBD-transactivator fusion protein, which manage translocation of TtCBD-transactivator into the nucleus to trigger expression of transgenes upon illumination. We show that Apple-Watch-programmed percutaneous remote control of implanted Glow-controlled engineered human cells can effectively treat experimental type-2 diabetes by producing and releasing human glucagon-like peptide-1 on demand. Directly interfacing wearable smart electronic devices with therapeutic gene expression will advance next-generation personalized therapies by linking biopharmaceutical interventions to the internet of things.

[1] Department of Biosystems Science and Engineering, ETH Zurich, Basel, Switzerland. [2] Faculty of Science, University of Basel, Basel, Switzerland.
✉email: fussenegger@bsse.ethz.ch

Programming cellular behavior using trigger-inducible gene switches is a central goal in synthetic biology-inspired approaches to deliver controlled dosages of therapeutic agents on demand, as well as to program the behavior of engineered cells in future cell-based therapies[1]. Many mammalian gene switches have been designed to sense and respond to stimuli such as chemical cues, disease metabolites, or biomarkers[2–4]. However, chemical-based inducers are often limited by side effects, bioavailability, or pharmacodynamics[4]. By contrast, light has been used for precise and traceless spatiotemporal remote control of cellular behavior and expression control of therapeutic transgenes in a variety of experimental disease models[5–10].

Currently available optogenetic mammalian gene switches or photoreceptors operating in the ultraviolet, blue, red, far-red, or near-infrared regions[11] exhibit various limitations that preclude their use in clinical applications. Nonlimiting examples include cytotoxicity[12,13], low tissue penetration[14], deregulation of endogenous genes[15], and the need for exotic chromophores that must be provided exogenously (e.g., the PhyB-PIF system[16]). Green-light optogenetics remains a "blind spot" in the clinical gene switch portfolio. CarH, a helix-turn-helix photoreceptor protecting *Thermus thermophilus* from cytotoxic effects caused by sunlight[17], has recently been adapted for mammalian cells using either activation of receptor signaling[18] or a topology blueprint of the pioneering tetracycline-repressible gene regulation system[19,20]. However, these systems are immediately and constitutively active upon transfection in the dark[18,20]. This requires continuous green-light illumination for expression tuning and shutdown, resulting in cytotoxicity that precludes application in disease models or potential clinical translation[18].

Smart wearable electronic devices such as the Apple Watch are equipped with green light-emitting diodes (LEDs) for photoplethysmography, to monitor health-related parameters, such as heart rate, heart-rate variability, electrocardiograms, atrial fibrillation, blood pressure, and sleep[21–24]. Apple Watch software enables the green LEDs to be operated in a user-defined or automatic manner, and the data can be transferred to healthcare professionals for analysis, either directly by a cellular-capable model or by connecting the watch to a cellular phone[25,26]. So far, the green-light-emitting functionality of smart wearable electronic devices has been used exclusively to record health parameters, but we considered that it might be feasible to use this functionality to achieve light-inducible percutaneous remote control of therapeutic protein production in subcutaneously implanted cells in a user-defined or automatic manner. To take this idea forward, we focus on the cobalamin-binding domain (CBD) of the CarH protein of *T. thermophilus* (TtCBD)[27], which assembles as a dimer of dimers in the dark when bound to the 5'-deoxyadenosylcobalamin chromophore, an active coenzyme form of vitamin B$_{12}$[28]. This tetrameric structure dissociates into monomers upon exposure to green light. We design a green-light-inducible gene switch topology that is induced by green-light illumination programmed by an Apple Watch and remains inactive when the watch's LED is switched off. Consequently, a synthetic transcription factor containing a TtCBD domain is by default sequestered by a supercharged TtCBD plasma-membrane anchor. Only upon green-light illumination does the TtCBDs dissociate and releases the transcription factor, which activates specific target promoters. We confirm that this green-light-operated smart-watch control system (Glow Control) provides robust, tunable, and reversible on-demand transgene expression in response to green-light illumination in various mammalian and human cell types. Furthermore, as a proof-of-concept, we show that subcutaneously implanted microencapsulated Glow-controlled engineered human cells can effectively treat diabetes and associated symptoms, including postprandial hyperglycemia, insulin resistance, fasting blood glucose levels, and obesity, in a mouse model of type-2 diabetes by means of percutaneously remote-controlled release of human glucagon-like peptide-1 (hGLP1) using programmed green-light illumination.

## Results

**Design of Glow-controlled mammalian cells**. To establish whether the TtCBD domains are compatible with smart-watch-triggered green-light-mediated dissociation, we designed a synthetic transcription factor by fusing one TtCBD domain C-terminally to VPR (VP64-p65-Rta; TtCBD-VPR) and a second TtCBD domain N-terminally to the tetracycline-dependent repressor (TetR; TetR-TtCBD). Upon co-transfection, TtCBD-TetR and TtCBD-VPR immediately heterodimerize to constitute the transcription factor TetR-TtCBD-TtCBD-VPR, which activates co-transfected expression units containing classical tetracycline-responsive promoters (P$_{TET}$), unless the transcription factor dissociates following illumination with green light (Supplementary Fig. 1a–c). However, this gene-switch configuration is incompatible with smart electronic wearable devices, as it would rapidly drain the battery owing to the constant illumination required to keep the therapeutic gene shut down. Wearable-controlled gene switches should remain shut down by default in unstimulated state and be specifically induced following illumination by the wearable's green-light LED.

To build our Glow Control, we designed a gene switch topology consisting of a plasma membrane-tethered myristylated TtCBD (Myristoylation signal peptide-TtCBD), and a P$_{TET}$-activating TtCBD-TetR-VPR transactivator (TtCBD-Transactivator; TtCBD-TA) (Fig. 1a, b) (see Supplementary Fig. 4 for VPR alternatives). In the un-illuminated state, the TtCBD moiety of the transactivator homodimerizes with its plasma membrane-tethered myristylated TtCBD counterpart, which sequesters the transactivator and prevents it from reaching and activating P$_{TET}$. Green-light exposure dissociates the TtCBD complex, releasing the transactivator, which in turns activates the target genes (Fig. 1c, d and Supplementary Figs. 2 and 3). Indeed, cells equipped with the Glow Control system showed high glycoprotein production in the presence of green light, in contrast to basal expression in the absence of illumination.

As TtCBD natively assembles as a homo-oligomer[28], Glow Control performance may be limited by *cis*-dimer formation (Myr-TtCBD/Myr-TtCBD or TA-TtCBD/TtCBD-TA), reducing the pool of necessary *trans*-dimers (Myr-TtCBD/TtCBD-TA). To increase the specificity of *trans*-dimer interaction, we equipped the membrane-anchored Myr-TtCBD and TtCBD-TA with positively or negatively super-charged domains in various combinations (Fig. 2a), which served to boost the overall Glow Control performance compared to non-supercharged controls or the original design (Fig. 2b and Supplementary Fig. 5). This performance increase is likely caused by a combination of electrostatic repulsion and steric hindrance (Fig. 2c). An improved gene switch composed of negatively charged plasma membrane anchored TtCBD (Myr-TtCBD-nGFP) and uncharged TtCBD-transactivator (TtCBD-TA) showed the lowest leakiness and a high level of fold induction (Fig. 2b, c and Supplementary Fig. 5) and was thus selected for all following experiments. This genetic circuit was also confirmed to be functional in various mammalian cell types, further broadening the range of potential applications (Fig. 2d).

**Characterization of the Glow Control cells**. Detailed characterization of the Glow Control system in human cells illuminated with green light showed significant induction of the

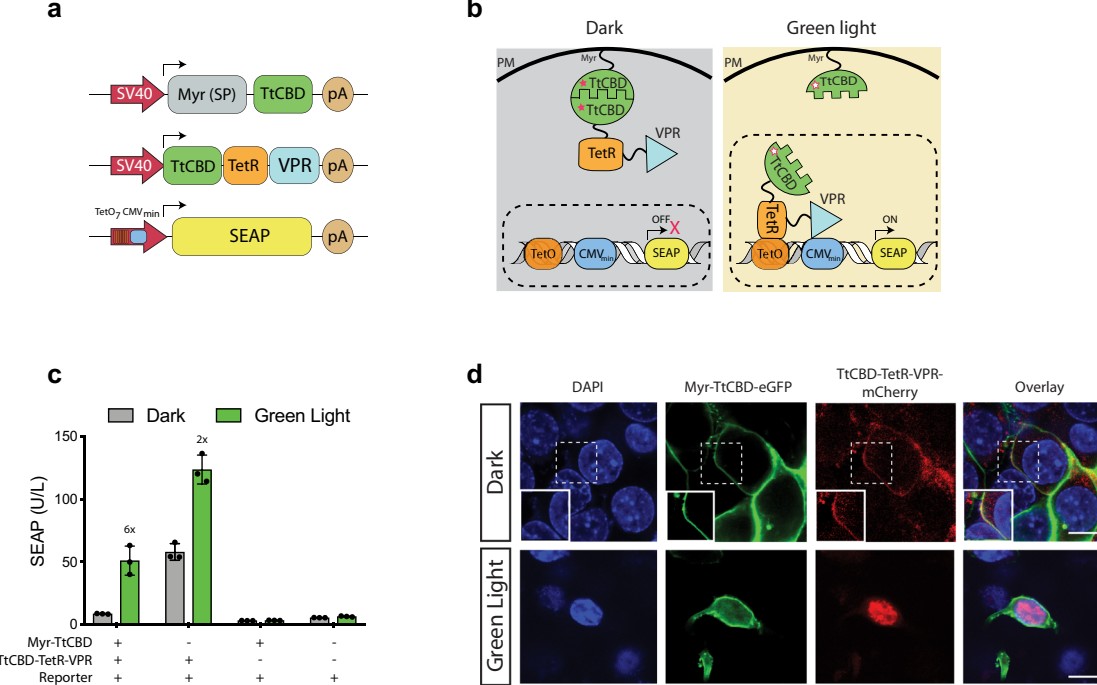

**Fig. 1 Design of the green-light-operated smart-watch control system (Glow Control) for mammalian cells. a** The DNA constructs used for the Glow Control system. The Glow Control circuit is based on dissociation of a Myristoylation signal peptide-TtCBD (pMMZ269; $P_{SV40}$-Myr-TtCBD-pA) and TtCBD-TetR-VPR transactivator (pMMZ273; $P_{SV40}$-TtCBD-TetR-VPR-pA), leading to expression of the SEAP reporter gene from the synthetic promoter $P_{TET}$ (pTS1017; $P_{TetO7}$-$P_{CMVmin}$-SEAP-pA). **b** Schematic representation of Glow Control in the dark and upon illumination with green light. In the presence of $AdoB_{12}$ (filled asterisk) and in the dark, TtCBD-TetR-VPR is bound to plasma-membrane-anchored Myristoylation signal peptide-TtCBD. Green light causes dissociation of the TtCBD-transactivator from the plasma membrane, which enables its translocation into the nucleus and subsequent activation of reporter gene (SEAP) expression. **c** Dependency of the Glow Control system on the presence (+) or absence (−) of each component was studied by quantification of SEAP in the supernatant of HEK293T cells transfected with the indicated plasmids. Cells were induced with pulsed green light (545 nm; 15 s ON/45 s OFF and 88 μW/cm²) for 48 h. Plots represent mean ± SD of $n = 3$ biologically independent samples. Numbers above the bars indicate fold changes of reporter (SEAP) vs. the corresponding dark group. **d** Microscopic images of HEK293T cells co-transfected with pMMZ351 ($P_{SV40}$-Myr-TtCBD-GFP-pA), pMMZ429 ($P_{SV40}$-TtCBD-TetR-VPR-mCherry-pA), and $P_{TET}$ (pTS1017; $P_{TetO7}$-$P_{CMVmin}$-SEAP-pA) in the dark and after pulsed green-light illumination (545 nm; 15 s ON and 45 s OFF and 88 μW/cm²) for 48 h. Cells were fixed and stained with DAPI. Scale bars are 10 μm. A representative image of three replicates from each group is shown. Source data are provided as a Source Data file.

reporter SEAP (human placental secreted alkaline phosphatase) after 6 h of green light illumination, followed by a time-dependent increase in SEAP levels (Fig. 3a). Illumination of Glow-controlled cells with green light for 12 h per day over the course of 4 days revealed robust and sustained induction profiles peaking on day 3 (Fig. 3b). Glow Control also showed excellent reversibility in response to cycles of green light exposure at 24 h intervals (Fig. 3c and Supplementary Fig. 6a, b). The half-life of the TtCBD-transactivator in Glow Control cells is about 12 h, which can be reduced to about 4 h by tagging the TtCBD-transactivator to a degron domain (Supplementary Fig. 6c, d). In addition, Glow Control can be turned off at any time simply by addition of the clinically licensed antibiotic doxycycline, which targets the TetR-domain and releases TtCBD-TA from $P_{TET}$ (Fig. 3d). Thus, doxycycline can be considered as a safety switch that is able to tune or shut down transgene expression at any time in emergency situations or after completion of therapy. Glow Control can also be precisely fine-tuned by programming light-related parameters, e.g., by varying the green light pulse lengths and intervals (Fig. 3e), as well as the illumination intensity (Fig. 3f), with minimal impact on cell viability, even at maximum light intensity (Fig. 3f). Glow Control was also fully compatible with red and near-infrared optogenetic gene switches (Fig. 4a). We further co-transfected cells with Glow Control, as well as a near-infrared-responsive optogenetic system[29] rewired to orthogonal expression units. This enables spectral multiplexing for differential gene

expression control and programming of complex cellular behavior (Fig. 4b, c).

For real-world application, we programmed illumination of Glow-controlled human cells with green light emitted by an Apple Watch (Fig. 5a), which resulted in a fivefold SEAP induction within 1 h compared to cell cultures covered by the same Apple Watch with the green LED switched off. Thereafter, SEAP expression further linearly increased with increasing duration of exposure (Fig. 5b). We also built a light-weight, custom-designed, open-source-programmable, smart-watch-mimicking, green-light LED patch that showed identical Glow-controlled gene expression to that obtained with the Apple Watch, suggesting that, in principle, any kind of green-light-emitting wearable device could also be used to drive Glow Control (see below Supplementary Figs. 8 and 10).

**Smart watch-programmed Glow Control cells for treatment of T2D mice.** We confirmed that the green light emitted by the Apple Watch could pass through a skin patch placed over a cell culture well, to induce Glow-controlled human cells (Supplementary Fig. 7). Therefore, we next subcutaneously implanted microencapsulated human cells engineered for Glow-controlled SEAP expression on the back of mice and either exposed them to a "green-light shower," a 48-LED array installed on top of the cage (Supplementary Fig. 8a, b), or placed our lightweight LED

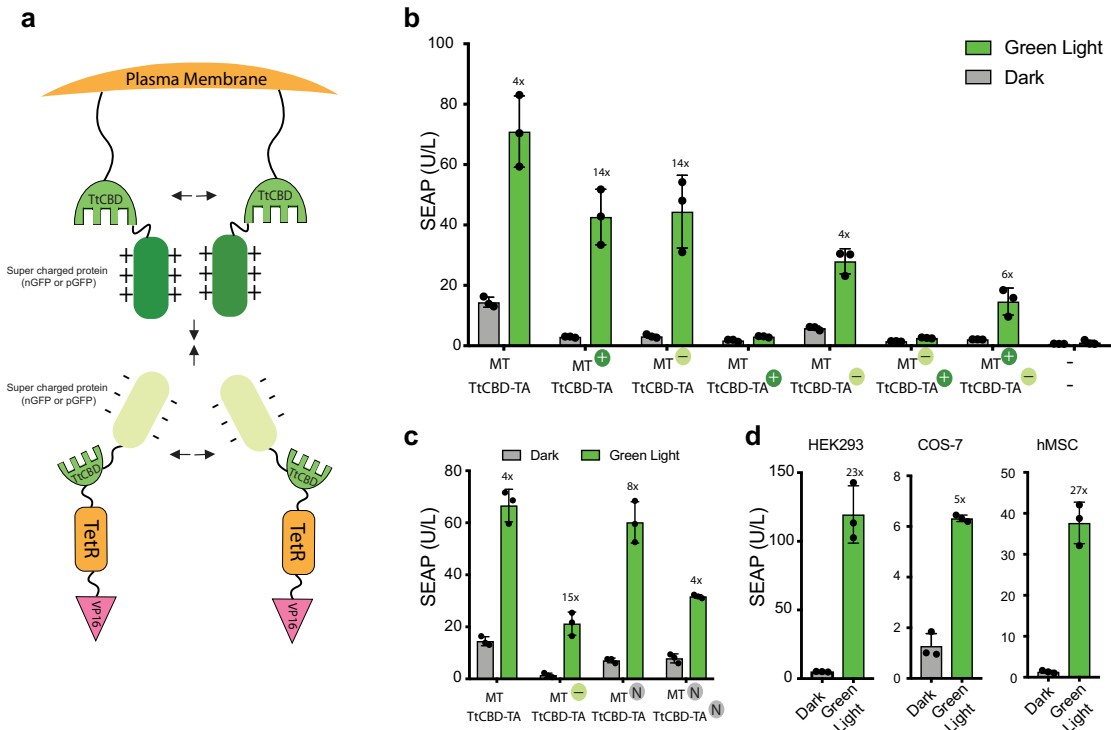

**Fig. 2 Improvement of Glow Control specificity. a** Schematic representation of supercharged GFP fusions to Glow Control components. Positively (+, pMMZ283; $P_{SV40}$-Myr-TtCBD-pGFP-pA) or negatively (−, pMMZ284; $P_{SV40}$-Myr-TtCBD-nGFP-pA) supercharged GFP was C-terminally fused to Myristoylation signal peptide-TtCBD (MT) and co-transfected in HEK293T cells with untagged (pMMZ272; $P_{SV40}$-TtCBD-TetR-VP16-pA) or N-terminally tagged with positive (+, pMMZ290; $P_{SV40}$-pGFP-TtCBD-TetR-VP16-pA) or negative (−, pMMZ291; $P_{SV40}$-nGFP-TtCBD-TetR-VP16-pA) supercharged GFP fused to the TtCBD-transactivator (TA) alongside a SEAP reporter construct (pT1017; $P_{TetO7}$-$P_{CMVmin}$-SEAP-pA). **b** Quantification of produced SEAP in the supernatant of HEK293T cells co-transfected with different variants (untagged, nGFP, or pGFP) of Myr-TtCBD (MT) and TtCBD-transactivator (TA). **c** Comparison of the efficiency of the optimized supercharged Glow Control with either untagged effectors or effectors tagged with native GFP (N, pMMZ351; $P_{SV40}$-Myr-TtCBD-GFP-pA, pMMZ309; $P_{SV40}$-GFP-TtCBD-TetR-VP16-pA) in a side-by-side experiment. **d** Validation of Glow Control-mediated SEAP expression in different mammalian cell lines. Cells were co-transfected with pMMZ284 ($P_{SV40}$-Myr-TtCBD-nGFP-pA), pMMZ272 ($P_{SV40}$-TtCBD-TetR-VP16-pA), and pT1017 ($P_{TetO7}$-$P_{CMVmin}$-SEAP-pA). For plots **b**–**d**, HEK293T cells (**b**, **c**) or indicated cells (**d**) were stimulated with pulsed green light (545 nm; 15 s ON/45 s OFF, and 88 μW/cm$^2$) for 12 h/day and SEAP expression was measured at 48 h after the start of illumination. Control cells were incubated in the dark. Values are mean ± SD ($n = 3$) and numbers above the bars indicate fold changes of reporter (SEAP) expression vs. the corresponding dark group. Source data are provided as a Source Data file.

patch (Supplementary Fig. 8c) or an Apple Watch on the animals' back above the implantation site (Fig. 5c). Interestingly, constant illumination for 3 h per day for 2 days by the Apple Watch and the LED patch led to similar SEAP expression levels in mice (Fig. 5d). It is noteworthy that the LED patch can easily be programmed to provide different light intensities and exposure times (Supplementary Figs. 10 and 11), and indeed produced even higher blood SEAP levels when programmed to give only 15 s of illumination per 1 min for 12 h per day, which is more energy-efficient than the Apple Watch's constitutive illumination (Fig. 5d).

To validate Glow Control in a clinical proof-of-concept study, we used the Apple Watch to percutaneously program the expression of the clinically licensed hGLP-1 for the treatment of experimental type-2 diabetes[10,30–34]. We assembled the transactivator components Myr-TtCBD-nGFP and TtCBD-TetR-VP16 on the same expression vector driven by different constitutive promoters to ensure optimal differential expression levels (Fig. 6a) and combined it with a $P_{TET}$-driven expression vector harboring a modified version of hGLP1[10,32] to produce a monoclonal human cell line Glow$_{hGLP1}$ (Fig. 6b and Supplementary Figs. 12 and 13). Type-2 diabetic mice subcutaneously implanted with microencapsulated Glow$_{hGLP1}$ exhibited increased circulating hGLP-1 levels (Fig. 6c) and showed attenuated

glycemic excursions in glucose-tolerance tests (Fig. 6d) upon illumination compared to the non-illuminated control group. In addition, Apple Watch Glow-controlled mice showed significantly lower fasting blood glucose levels (Fig. 6e), attenuated insulin resistance (Fig. 6f and Supplementary Fig. 14), and reduced body weight gain (Fig. 6g) compared to non-illuminated control groups over a 12-day treatment period. Throughout this period, Glow Control was reversible and explanted cells showed similar viability and performance to in-vitro-cultivated Glow$_{hGLP1}$ cells (Fig. 6h and Supplementary Fig. 15). These results validate the functionality of this system for the treatment of experimental type-2 diabetes.

## Discussion

The Glow Control system functionally connects optogenetic interventions to real-world wearable electronic devices and opens up the possibility of real-time remote programming of cell-based therapies for disease management[35] or lifestyle improvement[36,37]. In addition, self-learning artificial intelligence (AI)-based algorithms may control disease in an automatic and seamless manner, transforming medical interventions from the treatment of symptoms to curing disease[4,38]. This would be a dramatic expansion of existing applications of wearable devices[9,39], e.g., to

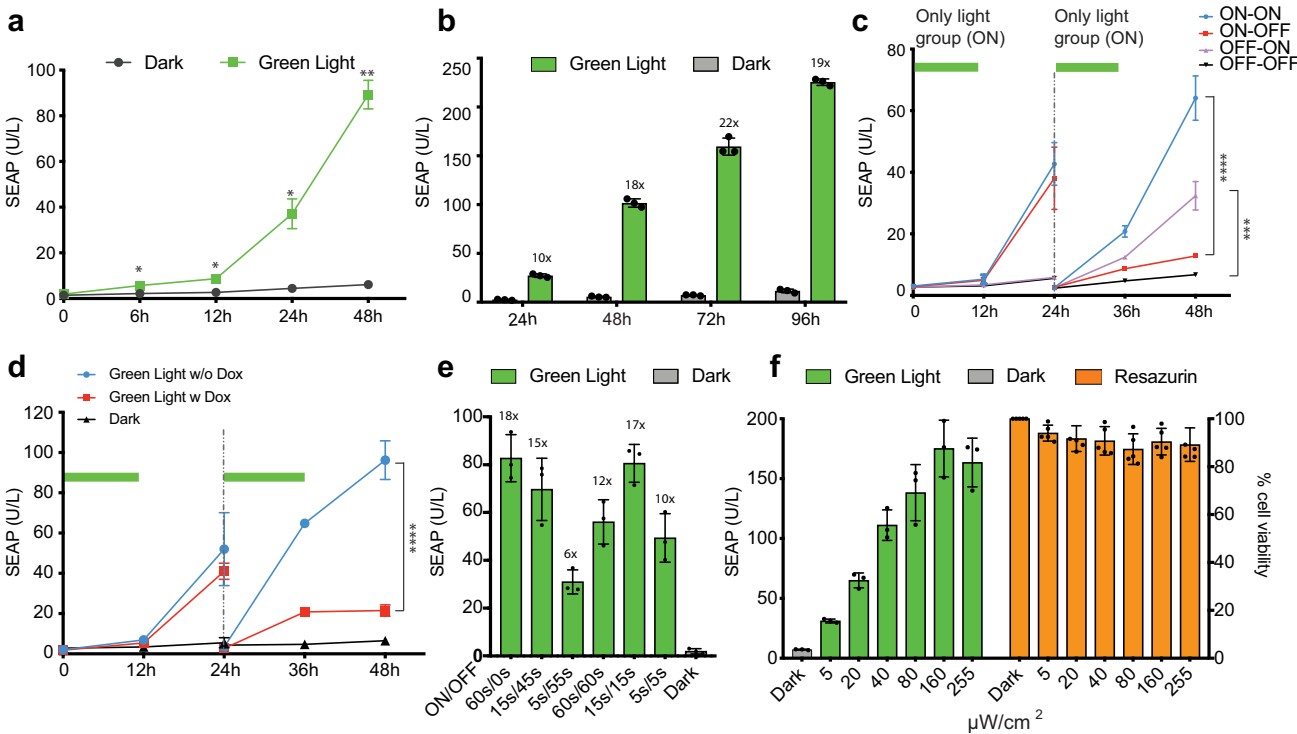

**Fig. 3 Characterization of Glow Control-transfected cells.** HEK293T cells were transiently co-transfected with pMMZ284 ($P_{SV40}$-Myr-TtCBD-nGFP-pA), pMMZ272 ($P_{SV40}$-TtCBD-TetR-VP16-pA), and pTS1017 ($P_{TetO7-CMVmin}$-SEAP-pA). Cells were illuminated with pulsed green light (545 nm; 15 s ON/45 s OFF, and 88 µW/cm²) for 12 h/day and SEAP levels were assessed in the supernatant of cultured cells at 48 h after the start of illumination, unless indicated otherwise. **a** In vitro kinetics of Glow Control system in response to green light. Cells were either exposed to pulsed green light or not illuminated. Values are mean ± SD ($n = 3$) and statistical significance between illuminated and un-illuminated groups were calculated using a two-tailed, paired Student's $t$-test. **b** SEAP levels in culture supernatant of HEK293T cells were profiled at the indicated time points after 12 h/day illumination with pulsed green light. Control cells were not illuminated. **c** Reversibility of the Glow Control system. HEK293T cells were illuminated with pulsed green light (ON) or were kept in dark (OFF) and SEAP levels were profiled every 12 h for 48 h. After 24 h, the culture medium was changed and the cell density was adjusted to $5 \times 10^4$ cells/well of a 96-well plate. Cells were again illuminated with (ON) or without (OFF) additional 12 h pulsed green light. The values are mean ± SD ($n = 3$). Statistical significance between indicated groups were calculated using a one-way ANOVA. **d** Doxycycline-controlled transcriptional inactivation of the Glow Control system. HEK 293T cells were illuminated for 12 h with pulsed green light. After 24 h, the medium was changed to medium with or without 1 µg/ml doxycycline and illuminated for another 12 h. The values are mean ± SD ($n = 3$). Statistical significance between indicated groups were calculated using a one-way ANOVA. **e** SEAP expression levels in HEK293T cells treated with green light pulses for various periods as indicated. **f** Light-intensity-dependent Glow Control activity and viability. HEK293T cells were illuminated with green light at different intensities (0–300 µW/cm², left). Viability of green-light-treated cells exposed to different light intensities (left) was assessed by MTT assay (right). Plots show the mean ± SD of $n = 3$ biologically independent samples. Numbers above the bars indicate fold changes of reporter (SEAP) expression vs. the corresponding Dark group. *$p < 0.05$; **$p < 0.01$; ***$p < 0.001$; ****$p < 0.0001$. Source data are provided as a Source Data file.

detect atrial fibrillation and deliver life-saving medical alerts[40,41]. Most current wearable biosensors[42] focus on data collection[43–46] or non-invasive real-time monitoring of physiological values, such as glucose levels in tears[47], perspiration[48], and interstitial fluids[49]. Furthermore, artificial-intelligence-driven algorithms programming non-invasive multi-sensor smart watches can profile a combination of heart rate, blood pressure, respiration, and oxygen saturation[50], to provide continuous blood-glucose monitoring[51], and are currently in U.S. Food and Drug Administration-approved clinical trials[52,53]. Nevertheless, these sophisticated monitoring devices lack direct feedback-controlled molecular intervention capabilities. We believe Glow Control technology provides the missing link between monitoring and intervention, using the same non-invasive, clinically licensed green-light LED interface and programming standards already employed in smart watches such as the Apple Watch. Bioelectronic interfaces established for mind-controlled gene expression[54], as well as for optogenetic[9] and electrogenetic[55] control of glucose homeostasis in experimental type-1 diabetes highlight the potential of electro-molecular therapeutic

interventions, but current devices require surgical implantation, as well as wireless power and control, which are associated with real-world challenges and risks. Glow Control, based on non-invasive wearable consumer electronics interfacing a standard programmable green-light LED with biopharmaceutical production of engineered cells, provides a new level of control, which we have successfully validated for experimental diabetes treatment. The combination of Glow Control with non-invasive, continuous glucose monitoring[47–49,51,52,56,57] could eventually enable full closed-loop control, a cell-based treatment strategy[4] that has been used to cure a variety of experimental metabolic disorders[32,58–60] or infectious diseases[61,62]. To translate our proof-of-concept smart watch-controlled Glow Control system into clinical reality, it will be necessary to develop improved cell encapsulation strategies, including novel approaches and materials that ensure long-term cell survival. In addition, immunogenicity of the bioengineered cells and their contents is a major safety barrier that restrict successful in vivo implementation[63]. Therefore, developing new strategies that can selectively modulate immune responses, such as the incorporation of immunomodulatory

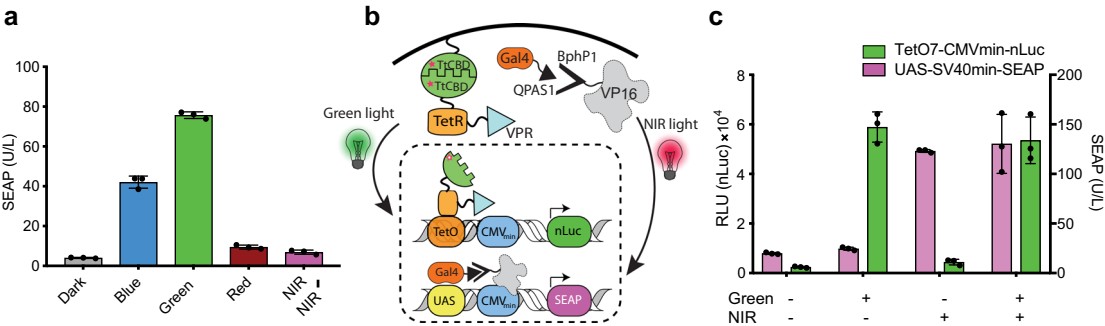

**Fig. 4 Function and application of Glow Control system at different wavelengths. a** Activation of the Glow Control system in response to light of different wavelengths. HEK293T cells transfected with pMMZ284 ($P_{SV40}$-Myr-TtCBD-nGFP-pA), pMMZ272 ($P_{SV40}$-TtCBD-TetR-VP16-pA), and pTS1017 ($P_{TetO7-CMVmin}$-SEAP-pA) were exposed to blue (475 nm, 88 µW/cm²), green (545 nm, 88 µW/cm²), red (660 nm, 8 µW/cm²), and NIR (740 nm, 1 mW/cm²) light. SEAP levels in the supernatant of cultured cells were profiled at 48 h after the start of illumination. **b** Spectral multiplexing of the Glow Control and BphP1-Q-PAS1 systems. HEK293T cells were transiently co-transfected with Glow Control constructs (pMMZ284, pMMZ272, and pLeo1403; $P_{TetO7-CMVmin}$-nLuc-Fc-pA) and BphP1-Q-PAS1 (pQP-T2A; $P_{CMV}$-BphP1-VP16-T2A-Gal4DBD-QPAS1-pA, and pSP30; $P_{UAS5}$-SEAP-pA) systems in a 1:1 ratio. **c** Cells co-transfected with Glow Control and BphP1-Q-PAS1 systems were incubated either without illumination, or illuminated with pulsed green light (545 nm, 12 h/day, 15 s ON/45 s OFF, 88 µW/cm²), NIR (740 nm, 24 h/day, 30 s ON/180 s OFF, 1 mW/cm²), or illuminated with both green light from the top and NIR from the bottom. The medium was supplemented with 10 µM B₁₂ and 25 µM biliverdin. SEAP and nLuc levels in the supernatant of cultured cells were measured at 48 h after the start of illumination. Plots show the mean ± SD of $n = 3$ biologically independent samples. Source data are provided as a Source Data file.

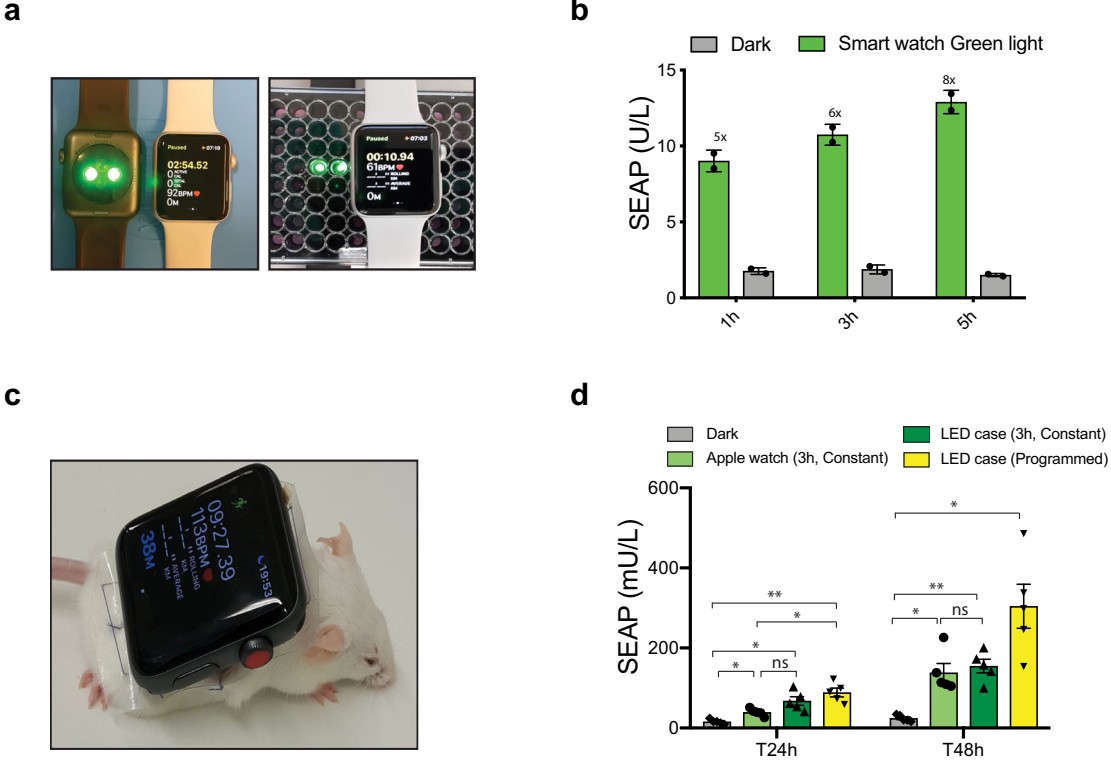

**Fig. 5 Smart-watch-mediated transgene expression in vitro and in vivo. a** Representative image of Apple watch-mediated in vitro gene expression. **b** HEK293T cells transfected with pMMZ284 ($P_{SV40}$-Myr-TtCBD-nGFP-pA), pMMZ272 ($P_{SV40}$-TtCBD-TetR-VP16-pA), and pTS1017 ($P_{TetO7-CMVmin}$-SEAP-pA) were exposed to green light emitted from an Apple Watch (0.15 mW/cm²) for increasing periods of time (1 h to 5 h per day). SEAP expression in cell culture supernatant was assayed at 48 h after the start of illumination. Plots show the mean ± SD of $n = 2$ biologically independent samples. **c** Representative image of RjOrl:SWISS (CD-1) mouse with an Apple Watch fixed on its back. **d** RjOrl:SWISS (CD-1) mice implanted with $10^7$ microencapsulated HEK293T cells (transiently co-transfected with pMMZ284/pMMZ272/pTS1017) and illuminated with green light emitted from an Apple Watch (3 h, constant, 150 µW/cm²) or a light-weight custom-built LED patch (either 3 h constant or pulsed for 12 h/day, 15 s ON/45 s OFF, 150 µW/cm²). SEAP levels in the blood were measured for 2 days. Experimental groups are indicated above the plot and values are mean ± SEM ($n = 5$). Statistical significance between indicated groups were calculated using a one-way ANOVA. ns, not significant; *$p < 0.05$; **$p < 0.01$. Source data are provided as a Source Data file.

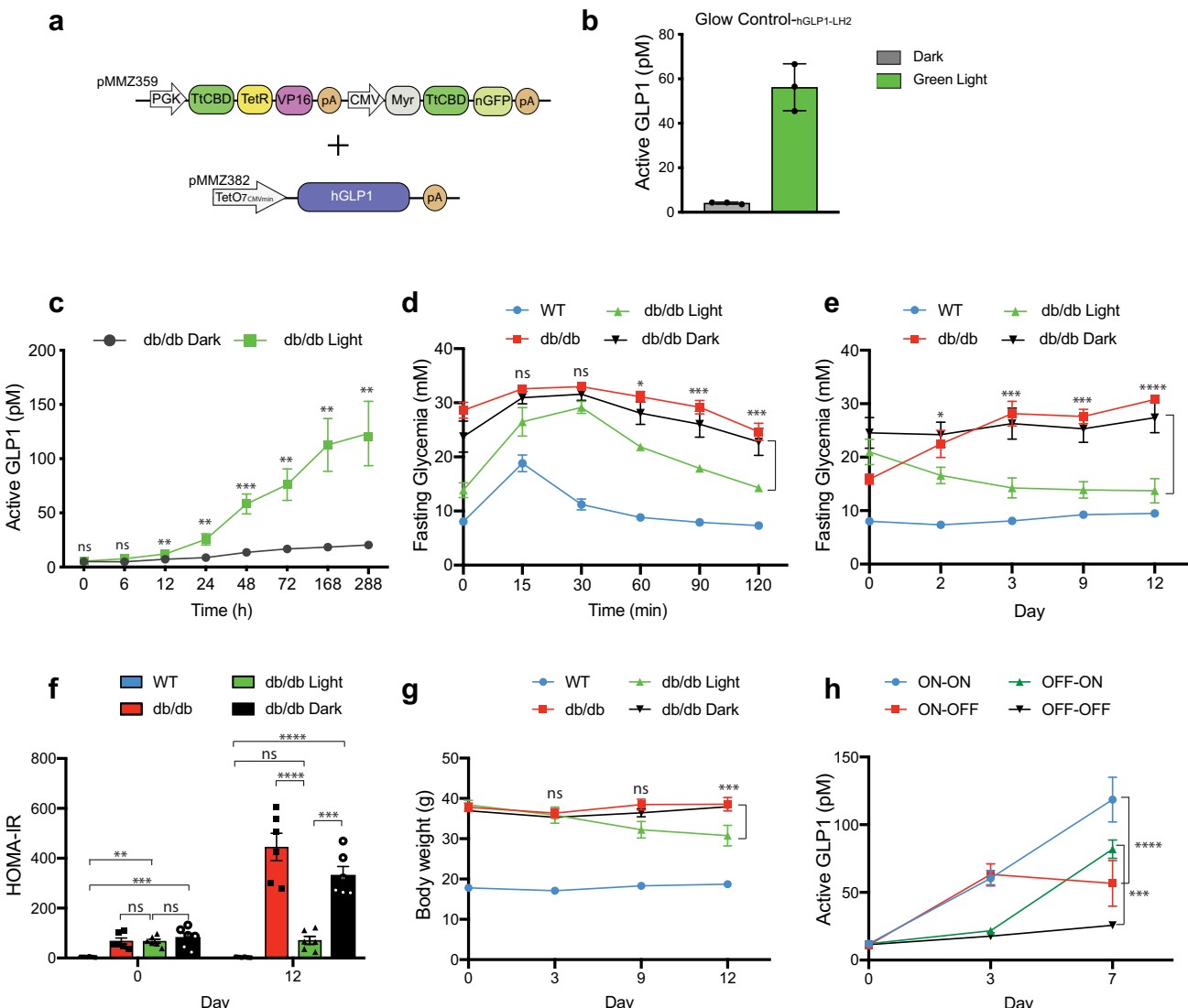

**Fig. 6 Validation of Glow Control for treatment of experimental type-2 diabetes. a** Schematic representation of HEK293T cells stably encoding a green-light-responsive module (pMMZ359; $P_{PGK}$-TtCBD-TetR-VP16-pA_$P_{CMV}$-Myr-TtCBD-nGFP-pA_$P_{RPBA}$-Ypet-p2a-Puro-pA) and a response module including a modified version of hGLP1 as a therapeutic gene (pMMZ382; $P_{TetO7-CMVmin}$-hGLP1-pA). **b** Green-light responsiveness of monoclonal Glow Control-$_{hGLP1}$ cells bearing the therapeutic cassette in vitro. Active GLP1 levels in the supernatant of cultured cells were profiled at 48 h after the start of illumination with green light (545 nm, 12 h/day, 30 s ON/30 s OFF, and 88 μW/cm²). Values are mean ± SD ($n = 3$). **c** Green-light-triggered Glow Control cell-based treatment of type 2 diabetic mice. Type-2 Lpr$^{db/db}$ mice (BKS-$Lepr^{db/db}$/JOrlRj) were subcutaneously implanted with $1 \times 10^7$ microencapsulated HEK293T Glow Control-$_{hGLP1}$ cells and illuminated with pulsed green light (545 nm, 12 h/day, 30 s ON/30 s OFF, 150 μW/cm²) from a custom-built, light-weight LED patch, or not illuminated. Blood GLP1 levels were measured 1 day after cell implantation before illumination ($T_0$) and over 12 days of illumination. Values are mean ± SD ($n = 5$) and statistical significance between illuminated and un-illuminated groups over 12 days were calculated using a two-tailed, paired Student's $t$-test. **d** Intraperitoneal glucose tolerance test (IPGTT) was performed by administration of 1.25 g/kg aqueous D-glucose and was conducted 72 h after the start of illumination (LED patch, 545 nm, 12 h/day, 30 s ON/30 s OFF, 150 μW/cm² ($n = 7$). Wild-type (WT) C57BL/6JRj mice ($n = 10$) and BKS-$Lepr^{db/db}$/JOrlRj ($db/db$, $n = 10$) without the cell implant were used as controls. Values are mean ± SEM and statistical significance between db/db illuminated and un-illuminated groups were calculated using a two-way ANOVA. **e** Twelve-day glycemic control in T2D mice implanted with Glow Control-$_{hGLP1}$ cells and illuminated with or without pulsed green light (LED case, 545 nm, 12 h/day, 30 s ON/30 s OFF, 150 μW/cm²) for 12 h daily. Values are mean ± SEM ($n = 6$) and statistical significance between db/db illuminated and un-illuminated groups were calculated using a two-way ANOVA. **f–h** Effect of green-light-triggered hGLP-1 expression on HOMA-IR (**f**) and body weight (**g**) of T2D mice during 12 days of pulsed illumination (LED case, 545 nm, 12 h/day, 30 s ON/30 s OFF, 150 μW/cm²). Values are mean ± SEM ($n = 6$) and statistical significance between indicated groups were calculated using a one-way ANOVA for **f** and a two-way ANOVA for **g**. **h** In vivo reversibility of the Glow Control-$_{hGLP1}$ cells in T2D mice in the presence (ON) or absence (OFF) of pulsed green light (LED patch, 545 nm, 12 h/day, 30 s ON/30 s OFF, 150 μW/cm²). Values are mean ± SEM for $n = 3$ T2D mice and statistical significance between indicated groups were calculated using a two-way ANOVA. ns, not significant; *$p < 0.05$; **$p < 0.01$; ***$p < 0.001$; ****$p < 0.0001$. Source data are provided as a Source Data file.

cytokines like chemokine (C-X-C motif) ligand (CXCL12) into microcapsules[64], will be needed. Alternatively, implementing the Glow Control genetic circuit in a clinically validated human cell type, e.g., patient-derived autologous somatic cells, would increase the level of safety. Furthermore, protection of the light-sensitive cells from ambient light to avoid any unprogrammed therapeutic gene expression is crucial for clinical application.

Compared with the artificial pancreas[65], which combines invasive continuous blood-glucose monitoring with real-time insulin release, a Glow Control-based system offers many advantages arising from its compatibility with green-light-LED-equipped smart wearable electronic devices that are programmable across different software ecosystems and already licensed for medical parameter monitoring.

## Methods

**Molecular cloning and DNA constructs**. Details of the design and construction of the expression vectors are provided in Supplementary Tables 1 and 2.

**Cell culture and transient transfection**. Human embryonic kidney cells (ATCC: CRL3216, HEK-293T), adipose tissue-derived human telomerase reverse transcriptase-immortalized human mesenchymal stem cells (ATCC: SCRC4000, hMSC-hTERT) and CV-1 (simian)-derived cells carrying SV40 (ATCC: CRL1651, COS-7) were cultured in Dulbecco's modified Eagle's medium (DMEM; Life Technologies, Carlsbad, CA, USA) supplemented with 10% (v/v) fetal bovine serum (FBS; Sigma-Aldrich, Munich, Germany) and 1% (v/v) penicillin/strepto-mycin solution (P/S: Sigma-Aldrich, Munich, Germany). All cells were cultured in a humidified atmosphere containing 5% $CO_2$ at 37 °C. Cell viability and number were assessed with an electric field multi-channel cell-counting device (CASY Cell Counter and Analyzer Model TT; Roche Diagnostics GmbH, Basel, Switzerland).

For transfection in a 24-well plate format, 500 ng of plasmid DNA was diluted in 200 μL FBS-free DMEM, mixed with 6.25 μL polyethyleneimine (PEI; Polysciences Inc.; 1 mg mL$^{-1}$), and incubated at room temperature for 20 min. Then, the transfection mixture was added to $6.3 \times 10^4$ cells, which had been seeded 12 h earlier. For each construct, three wells in a 24-well plate were used. Six hours after transfection, the transfected cells were trypsinized, pooled, and resuspended in a total volume of 800 μL of complete medium containing 10 μM coenzyme B$_{12}$ (cat. No. C0884, Sigma-Aldrich). 120 μL of the cell suspension was seeded in each well of a poly-L-lysine-pretreated 96-well plate. Three wells in a 96-well plate were considered as a Light group and three wells in another unilluminated 96-well plate were considered as a Dark group (120 μL × 3 for Light + 120 μL × 3 for Dark). Twelve hours after re-seeding, the medium was replaced by DMEM without Phenol red, supplemented with 10% FBS, 1% P/S and 10 μM coenzyme B$_{12}$. Cells were either illuminated from an LED array placed above the plate or kept in the dark. Transgene expression was evaluated at 48 h after the start of illumination.

**Light experiments**. We constructed a 96-LED array platform with multiple tunable parameters (e.g., light intensity, irradiation pattern) for illumination of individual wells in 96-well microwell plates for in vitro experiments. Details of its construction as well as its Arduino programming set-up are provided in Supplementary Figure 9 and Supplementary Note 1 Cells were irradiated with green LEDs (545 nm, LED545-01), blue LEDs (475 nm, B56L5111P), red LEDs (660 nm, B5B-436-30) and near-infrared LEDs (740 nm, LED740-01AU). All the LEDs were purchased from Roithner LaserTechnik, Austria. For in vivo experiments, we used either smart watches (Apple, series 3, 42 mm) or custom-built, wirelessly powered, smart-watch-mimicking LED patches with the same dimensions as the smart watch, but lighter in weight. For details, see Supplementary Figures 10 and 11. In the Apple watch experiment, green light illumination was applied via the "Running outdoor" app on the watch.

**SEAP assay**. For the quantification of human placental-SEAP, the cell culture supernatant was heat-inactivated for 30 min at 65 °C. Then, 10 μL of supernatant was diluted with 90 μL dH$_2$O and mixed with 80 μL 2 × SEAP buffer (20 mM homoarginine, 1 mM MgCl$_2$, 21% (v/v) diethanolamine, pH 9.8) and 20 μL of substrate solution containing 20 mM pNPP (Acros Organics BVBA). Absorbance at 405 nm was measured using a Tecan microplate reader (TECAN AG, Maen-nedorf, Switzerland). SEAP production in vivo was quantified with the chemilu-minescence SEAP reporter gene assay (cat. no. 11779842001, Sigma-Aldrich) according to the manufacturer's instructions. Background SEAP values were not subtracted.

**Microscopy**. For microscopic analysis, cells were plated in black 96-well plates, F-bottom (Greiner bio one, lot # E18053JY) treated with poly-L-lysine (Sigma P4707). Analysis was performed after 48 h of green-light illumination. Cells were fixed with 4% formaldehyde in PBS and stained with DAPI. Imaging was

performed on a Leica SP8 laser scanning confocal microscope. DAPI was excited with the 405 nm laser line and emission was collected from 430 to 450 nm (405/430-450). The other fluorescent proteins were analyzed as follows: GFP (514/525-545), mCherry (543/585-620).

**Generation of stable cell lines**. To develop stable therapeutic Glow Control cell lines according to the Sleeping Beauty transposon protocol[66], one well of a 6-well plate containing HEK-293T cells was co-transfected with pMMZ359 (475 ng) and pMMZ382 (475 ng) as well as 50 ng of the Sleeping Beauty transposase expression vector pTS395 (P$_{hCMV}$-SB100X-pA). After 12 h, the transfection medium was exchanged to culture medium supplemented with 10% FBS, 1 %P/S, and 10 μM Coenzyme B$_{12}$. After an additional 24 h, the medium was exchanged for standard culture medium supplemented with 2.5 μg/mL puromycin (catalog number ant-pr-1; Invivogen) and 5 μg/mL blasticidin (catalog number ant-bl-1; Invivogen) and a polyclonal cell population was selected after 1 week. Cells were sorted by means of fluorescent-activated cell sorting (FACS) into four different subpopulations according to their Ypet and BFP fluorescence intensities (High/High, High/Low, Low/High, and Low/Low), and each individual sorted cell was grown in a well of a 96-well plate. FACS-related data was analyzed using FlowJo 10.5 software. Monoclonal cell populations were screened for green-light-responsive GLP1 expression. Glow Control$_{GLP1}$was selected as the best performer and was used for follow-up experiments.

**Analytical assays**. Blood levels of GLP-1 in tested mice were measured with the High Sensitivity GLP-1 Active ELISA Kit, Chemiluminescent (Sigma-Aldrich; catalog number EZGLPHS-35K), according to the manufacturer's instructions. NanoLuc levels in cell culture supernatants were quantified using the Nano-Glo Luciferase Assay System (N1110, Promega). For the glucose tolerance test, mice were challenged by intraperitoneal injection of glucose (1.25 g/kg body weight in H$_2$O) and the glycemic profiles were obtained by measurement of blood glucose levels with a glucometer (Contour® Next; Bayer HealthCare, Leverkusen, Germany) every 15 or 30 min for 120 min. The cytotoxicity of light to illuminated cells was monitored by MTT (3-(4,5-dimethylthiazol-2-yl)-2,5-diphenyltetrazolium bromide) assay (Sigma-Aldrich, Munich, Germany; catalog number M5655). After 48 h illumination of cells with green light, the medium was replaced with culture medium containing 20 mM resazurin and the cells were incubated for 2–4 h. The absorbance of resazurin was quantified optically with excitation at 570 nm and emission at 590 nm on a Tecan microplate reader. Mouse insulin in mouse serum was quantified with an Ultrasensitive Mouse Insulin ELISA kit (Mercodia; catalog number 10-1249-01). Homeostatic model assessment (HOMA) IR index, a measure of insulin resistance, was calculated according to the formula: HOMA-IR = [fasting glucose (mmol/L) × fasting insulin (mU/L)]/22.5[67].

**Western blotting**. HEK-293T cells were transfected in 10 cm plates (6 h), reseeded into six-well plates and grown overnight. Cell were treated without ($T_0$) or with 25 μg/ml cycloheximide (Sigma) in complete media, collected after 4, 8, and 12 h, and lysed with RIPA buffer (50 mM Tris pH 7.6, 150 mM NaCl, 1% Triton X-100, 1% Na-deoxycholate, 0.1% SDS) containing freshly added complete protease inhibitor cocktail (Roche) and 0.5 μl Benzonase® endonuclease (Novagen). After centrifugation for 10 min at 18,000 × g and 4 °C, the protein concentration of the supernatant was quantified using a BCA assay (ThermoFisher) and an aliquot containing 10 μg protein was run on a Bolt 4–12% Bis-Tris gel (ThermoFisher). Proteins were transferred to nitrocellulose membranes (Amersham) according to standard procedures. Mouse anti-HA (Sigma, H3663; diluted 1 : 10,000 in 5% non-fat dried milk/PBST) and rabbit anti-GAPDH (Abcam, ab9485; diluted 1 : 2500 in 5% non-fat dried milk/PBST) were used as primary antibodies. As secondary antibodies, alkaline phosphatase-coupled ECL$^{TM}$ anti-mouse and anti-rabbit IgGs (GE Healthcare, diluted 1 : 10'000 in 5% non-fat dried milk/PBST) were used, followed by chemiluminescence detection. Images were acquired using a Fusion FX apparatus (Vilbert Lourmat). Quantification was performed with ImageJ. Original western blottings are shown in the Data source file.

**Cell encapsulation**. Encapsulated Glow Control designer cells were generated with an Inotech Encapsulator Research Unit IE-50R (EncapBioSystems, Inc., Greifensee, Switzerland). Coherent alginate-poly-(L-lysine) beads (400 μm diameter, 500 cells per capsule) were generated with the following parameters: 200 μm nozzle with a vibration frequency of 1000 Hz; 25 mL syringe operated at a flow rate of 410 units; 1.2 kV bead dispersion voltage.

**Animal study**. All experiments involving animals were performed in accordance with the Swiss animal welfare legislation and approved by the veterinary office of the Canton Basel-Stadt (approval number 2879/31996).

**Experimental animals**. Male and female diabetic BKS-Lepr$^{db/db}$/JOrlRj and wild-type control C57BL/6JRj mice and RjOrl:SWISS (CD-1), aged 8–9 weeks, were used. Mice were obtained from Janvier Labs (Saint-Berthevin, France) and acclimatized for at least 1 week. Animals were housed with an inverse 12 h day–night cycle in a temperature (21 ± 2°C) and humidity (55 ± 10%)-controlled room with

ad libitum access to standard diet and drinking water supplemented with $10\,\mu M$ coenzyme $B_{12}$ (catalog number C0884, Sigma-Aldrich). Animals were randomly assigned to experimental groups.

**Implantation experiments**. Implantation was performed by subcutaneous injection of 1 mL glucose-free DMEM containing alginate-poly(L-lysine)-alginate-microencapsulated Glow Control designer cells in the shaved back on one side of the mice. One day after implantation, either custom-built LED patches or smart watches were applied at the site of injection and fixed in place with sutures. Plasma was collected by centrifugation (10 min, $5000 \times g$) of clotted blood (20 min at 37 °C and then 20 min at 4 °C).

**Statistical analysis**. Statistical tests and significance are reported in the figures and corresponding figure legends. A two-tailed, paired Student's $t$-test and one-way or two-way analysis of variance with Tukey's test were applied to determine the statistical significance of differences among groups, using GraphPad Prism.

**Reporting summary**. Further information on research design is available in the Nature Research Reporting Summary linked to this article.

## Data availability

The authors declare that all data supporting the findings of this study are available within the paper and its Supplementary Information files. Sequencing data of Glow Control plasmids have been deposited in GenBank under accession codes (pMMZ284; MW731449, pMMZ272; MW731450 and pTS1017; MT267334.1). All vector information is provided in Supplementary Table 1. Requests for materials should be made to the corresponding author. All plasmids generated in this study are available upon request. Any additional relevant data are available from the authors upon reasonable request. Source data are provided with this paper.

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

## Acknowledgements

Work in the laboratory of M.F. is financially supported in part through a European Research Council advanced grant (ElectroGene, number 785800) and in part by the National Centre of Competence in Research (NCCR) for Molecular Systems Engineering, as well as the EC Horizon 2020 Framework Programme ENLIGHT. We thank Leo Scheller for helpful discussions, Henryk Zulewski for generous medical advice, and Emanuel Lörtscher for critical comments on wearable devices and electronics.

## Author contributions

M.M. and M.F. designed the project. M.M. and T.S. conducted in vitro experiments. G.C. designed the animal experiments. M.D.H., M.M., and S.X. performed animal experiments. P.B. and M.M. constructed and programmed the LED panels. M.M. and M.F. analyzed the data and wrote the manuscript.

## Competing interests

The authors declare no competing interests.
