## [Peer Review File · Nature Communications]

Reviewers' Comments:

Reviewer #1:

Remarks to the Author:

In the present publication Mansouri et al. present a novel optogenetic switch for the induction of gene expression using the green light switch TtCBD from *Thermus thermophilus*. Additionally, they use for the first time an Apple Smart Watch as illumination device. Doing so they want to make the system usable for "real-live applications". They probe the system in cell culture as well as in vivo. They show that activation of expression of the human glucagon-like peptide 1 (hGFP) which is used for the treatment of type-2 diabetes, is possible by green light illumination through the skin using an App of the commercially available Apple Smart Watch. The authors claim that "Glow control" is the "missing link between monitoring (health parameters) and intervention" for treatment of diseases like diabetes using Smart Watches. Such work might pioneer a new generation of smart wearables which cannot only sense (current wearables) but also control health parameters. The manuscript is very likely of broad interest.

However, in order to better be able to evaluate the performance of the system, it is recommended that the authors address the following points:

Major points:

1. How was the cofactor of TtCBD administered to the mice and does it have any side effects? Is the cofactor present in mammals and if yes at which concentration? If not: would it be necessary to administer the cofactor regularly to humans? This could strongly compromise the practical application of the system.
2. Figure 2b the abbreviation "MT" is nowhere explained. Furthermore, the system seems to work best when only "MT" is charged. Shouldn't opposite charges on "MT" and VP16 work best as they attract each other like it is schematically shown in Fig 2a? Why is this not the case? What was the rationale behind fusing the negatively charged protein to TetR-VP16? Isn't it expected that a negatively charged protein would impair interaction of this transcription factor with negatively charged DNA? It is recommended that the authors swap the charges: negative charge to the membrane-bound protein and positive charge to TetR-VP16 and test for better performance.
3. Figure 3c is very unclear. The green bars in the top of the figure seems to indicate that all samples were illuminated for 12 h at each day but the legend says: OFF-ON and OFF-OFF for two samples. Please clarify how the samples were illuminated. Furthermore, if my interpretation is right: purple (OFF-ON) and black (OFF-OFF) were kept in the dark for 12 h after change of medium. Shouldn't they be identical at $t = 36$ h and then diverge? As the purple sample increases a lot in the dark wouldn't that indicate a high leakiness of the system? Same for blue (ON-ON) and red (ON-OFF) I interpret that after change of medium those samples were first illuminated for 12 h and then "red" was put back to darkness. Why isn't the red signal at $t = 36$ h as high as the blue signal and flatten afterwards?
In conclusion, I cannot tell whether this system is reversible from this figure. Please clarify. Further: Given the irreversibility of light-mediated CarH dissociation (photolysis of the cofactor) it would be expected that the system will remain in the on-state as long as TtCBD-TetR-VP16 is present in the cell and that the system will only return to background levels once this protein has been degraded. Hence, the stability of TtCBD-TetR-VP16 seems to be the limiting factor in the reversibility of the system. In order to better quantify this process it is requested to experimentally determine the degradation rate of this protein.
4. How long do cells survive/remain functional in the microcapsules? Do they lose activity over time and would there be the need to transplant them regularly? How could this be done practically? Repeated surgery or other less invasive measures? Please provide a perspective for a real-world application in the discussion.

Minor points

1. Figure 2a shows the system with C-terminally tagged TtCBD (pMMZ284) which was also used for further experiments. Anyhow in Figure S5 it is shown that the N-terminally tagged version (pMMZ410) works better. Why was the C-terminally tagged version used then?

2. Figure 2c the "N" for the native GFP is not readable. All in all, a bigger font in the Figures would make reading more comfortable.
3. Figure 3a and b: Why is the fold change at 24 h smaller than after 12 h and why is this shown in 2 different experiments? Wouldn't make one continuous experiment more sense?
4. Figure 3h is too small and therefore not readable.
5. Supplementary Figure 2b: It looks like the TtCBD-TetR-VPR-mCherry construct shuttles into the nucleus upon green light exposure. How is that to be explained? Is there an NLS sequence in the protein?

Reviewer #2:

Remarks to the Author:

In this work, the authors engineered mammalian cells to respond to green light and in-turn induce the expression of transgenes. The application of this green light-inducible switch was demonstrated in diabetic mice where human GLP-1 expression was triggered by the green light from wearable electronic devices. Overall, the manuscript is reasonably well-written and the data is well-presented.

Major comments

1. For SEAP quantification assays, was there any normalization done (e.g. cell number)? If so, please specify.
2. In figure 2b: Very low levels of SEAP were observed for MT-VP16+, MT- -VP16+, MT+- VP16- when the green light was pulsed during the assay. Please comment on the reasons which may have led to this observation.
3. In figure 2d: Please state which plasmid constructs were used for the various cell lines.
4. In figure 2: Based on the results, please provide a detailed explanation of the specific choice of pMMZ284 (PSV40-Myr-TtCBD-nGFP-pA) which was employed for subsequent experiments.
5. In figures 3c and 3d: The reversibility of the system is not clearly demonstrated. For instance, the reason for reseeding cells at 24h and a subsequent difference in response after the cells were reseeded, is not clear. Also, how is reversibility demonstrated in these figures based on the suitability of SEAP as a reporter? Please address these concerns with detailed explanations of the observed data.
 - a. Please provide experimental data with statistical analysis showing reversibility of the system without reseeding of cells.
 - b. Please provide experimental data with statistical analysis showing reversibility with doxycycline up to 48h instead of 42h.
6. In figure 3e: Please define and provide details on how this result demonstrates the targeted level of precise fine-tuning.
7. In figure 3h: The design and the constructs used to achieve the data shown are not clear. Please provide a more in-depth analysis of the same in the text.
8. In Figure 4d, active hGLP-1 levels were measured for Glow-controlhGLP-1 cells with the illumination schedule 545nm, 12h/day, 15s ON/45s OFF. This illumination schedule differs from the subsequent in vivo experiments where the illumination pattern was 30s ON/30s OFF. Please provide relevant in vitro experimental data for active hGLP-1 levels from Glow-controlhGLP-1 cells

with illumination pattern of 30s ON/30s OFF.

9. In figures 4e-4i: Statistical significance between the groups is not well indicated. Please provide statistical analysis on the results and indicate significance in these figures.

10. In figure 4e: Blood hGLP1 levels were measured at Day 1 after cell implantation before illumination (T0) and at 48 h (T48) after the start of illumination. To further understand the time course of the hGLP1 level, provide the hGLP1 levels at Day 1 and 3 after the start of illumination.

11. In vitro data demonstrating reversibility of Glow-controlhGLP-1 cells is lacking to support Figure 4j. Please provide relevant experimental data with statistical analysis.

12. In order to provide a clear understanding of the translational capability of this technology, please discuss its advantages in the conclusion section. For instance, as the bioengineered cells need to be implanted under the skin:

- What is the half-life of the implanted cells? If it needs to be constantly re-implanted? If so, how often is it required?
- What are the safety issues involved in implanting these bioengineered cells under the skin?
- Upon switching on the light, how long does it take before a significant level of GLP1 can be detected in the blood stream?

Minor comments

1. Please include all plasmids used and designed in the supplementary table 1. pMMZ429 was not found in the table.

2. There is a repeat pMMZ351 listed found in the supplementary table 1. Please revise this appropriately.

3. In figure 1c: It shows that SEAP level (Myr-TtCBD-, TtCBD-TetR-VPR+, Reporter+) under green light is higher (2 ×) than that in Dark. Please explain the reasons for the difference in SEAP level.

4. In References, appropriate and consistent abbreviations of journal names (e.g. Trends in Biotechnology, Nature Reviews Molecular Cell Biology) should be used.

5. Throughout the manuscript, appropriate and consistent units such as days, hours, seconds (page 4, page 10) should be used.

Reviewer #3:

Remarks to the Author:

The authors have constructed a novel system using TtCBD, where optogenetic stimulation of genetically engineered cells induces a transgene expression. They have validated the system composed of genetically engineered hGLP-1 producing cells implanted subcutaneously in an diabetic mouse model and a green-light-emitting device including Apple Watch, showing robust improvements in their metabolic parameters.

First of all, the results presented here seem to be another version of what was already reported by the authors (Science, 2011). While the system reported here might provide novelty in terms of handiness for clinical use as illustrated by the use of Apple Watch, the reviewer feels that it offers limited advantages over its precedent version as well as other currently available therapeutic tools such as islet transplantation or sensor-augmented put therapy (SAP).

The potential limitations and other concerns are as following:

1, In the diabetic mouse model treated with the system, what is the kinetics of serum hGLP1 concentration over 12 days ?

2, Related to 2, as deduced from Supplementary Fig. 14, the cell number is speculated to fluctuate over time due to replication or apoptosis. This fluctuation would not only limit the practical durability of this system but also lead to instability of the transgene expression, leading to substantial problems. Could the authors overcome this issue ?

3, Given the wavelengths used in the experiments, it could be possible that the engineered cells might express the transgene without green-light stimulation when exposed to illumination or sunlight. The authors should discuss this issue.

4, The detailed description of hGLP1 is lacking. Is it a modified version so that it would not be cleaved by DPP-IV ? In that case, is the ELISA kit able to detect the modified GLP-1 ?

5, Related to 4, if the authors are using naive human GLP-1, the ELISA kit for GLP1 should not be specific for this transgene, as it detects naive GLP-1 derived from multiple species. In that case, the Y axis title of Fig 4d and 4e as well as the relevant texts should be amended.

6, In Fig4b and 4h, the comparison should be performed using One-way ANOVA with post hoc analyses.

7, In Fig4f, 4g, 4i and 4j, the comparison should be performed using ideally Two-way ANOVA (or alternatively One-way ANOVA) with post hoc analyses.

Revision of NCOMMS-20-43170-T – Point-By-Point Response

We are very grateful to all the referees for their constructive and useful criticisms. We have addressed all the comments in the revised version of our manuscript, and our point-by-point responses are given below.

Reviewer #1 Remarks to Author:

In the present publication Mansouri et al. present a novel optogenetic switch for the induction of gene expression using the green light switch TtCBD from *Thermus thermophilus*. Additionally, they use for the first time an Apple Smart Watch as illumination device. Doing so they want to make the system usable for “real-live applications”. They probe the system in cell culture as well as in vivo. They show that activation of expression of the human glucagon-like peptide 1 (hGFP) which is used for the treatment of type-2 diabetes, is possible by green light illumination through the skin using an App of the commercially available Apple Smart Watch. The authors claim that “Glow control” is the “missing link between monitoring (health parameters) and intervention” for treatment of diseases like diabetes using Smart Watches. Such work might pioneer a new generation of smart wearables which cannot only sense (current wearables) but also control health parameters. The manuscript is very likely of broad interest.

However, in order to better be able to evaluate the performance of the system, it is recommended that the authors address the following points:

We thank the reviewer for the positive assessment of our manuscript and in particular for the valuable suggestions for improving the clarity and presentation of the results. Our responses are as follows.

Major points:

1. How was the cofactor of TtCBD administered to the mice and does it have any side effects? Is the cofactor present in mammals and if yes at which concentration? If not: would it be necessary to administer the cofactor regularly to humans? This could strongly compromise the practical application of the system.

The Glow Control system relies on the abundant co-factor vitamin B12, an essential compound for humans (deficiency in vitamin B12 can lead to anemia and other diseases in humans (O’Leary and Samman, 2010)). B12 is a water-soluble vitamin, and is widely available as a dietary supplement and a prescription medication. There is no Tolerable Upper Intake Level (UL) for B12, due to its low level of toxicity even at high doses (<https://ods.od.nih.gov/factsheets/VitaminB12-HealthProfessional/#en5>). We supplemented cell media and animal water bottles with vitamin B12 (10 μ M) in both in vitro and in vivo experiments and have mentioned this in the methods section (pages 8 and 11).

2. Figure 2b the abbreviation “MT” is nowhere explained. Furthermore, the system seems to work best when only “MT” is charged. Shouldn’t opposite charges on “MT” and VP16 work best as they attract each other like it is schematically shown in Fig 2a? Why is this not the case? What was the rationale behind fusing the negatively charged protein to TetR-VP16? Isn’t it expected that a negatively charged protein would impair interaction of this transcription factor with negatively charged DNA? It is recommended that

the authors swap the charges: negative charge to the membrane-bound protein and positive charge to TetR-VP16 and test for better performance.

MT stands for plasma membrane tethered-TtCBD (**Myristoylation signal peptide-TtCBD; MT**). We have added “**Myristoylation signal peptide-TtCBD (MT)**” in both the text (page 3) and figure legend (Figures 1 and 2). The abbreviation **TA** is used to describe TtCBD-transactivator (TtCBD-TA) in the text (page 3) and figure legend 2.

Regarding fusions of supercharged GFPs to MT and TA, we first generated a version consistent with the reviewer’s comment in Figure 2b. We indeed expected that a magnet strategy based on oppositely charged MT and TA (i.e., MT^+/TA^- or MT^-/TA^+) would have the best performance. However, the results in Figure 2b show that tagging TtCBD-transactivator (TtCBD-TA) with a supercharged GFP can reduce its functionality. This might be due to steric hindrance of complex formation and/or hampered translocation of TtCBD-TA to the nucleus due to the larger size or altered charge of the complex (Colwell et al., “Charge as a Selection Criterion for Translocation through the Nuclear Pore Complex”, 2010).

We have already studied all the possible combinations including uncharged MT (MT), positively charged MT (MT^+) and negatively charged MT (MT^-) with uncharged TA (TA), positively charged TA (TA^+) and negatively charged TA (TA^-) (please see figure 2b). We have clarified this in the text (page 4).

3. Figure 3c is very unclear. The green bars in the top of the figure seems to indicate that all samples were illuminated for 12 h at each day but the legend says: OFF-ON and OFF-OFF for two samples. Please clarify how the samples were illuminated. Furthermore, if my interpretation is right: purple (OFF-ON) and black (OFF-OFF) were kept in the dark for 12 h after change of medium. Shouldn’t they be identical at $t = 36$ h and then diverge? As the purple sample increases a lot in the dark wouldn’t that indicate a high leakiness of the system? Same for blue (ON-ON) and red (ON-OFF) I interpret that after change of medium those samples were first illuminated for 12 h and then “red” was put back to darkness. Why isn’t the red signal at $t = 36$ h as high as the blue signal and flatten afterwards? In conclusion, I cannot tell whether this system is reversible from this figure. Please clarify.

Further: Given the irreversibility of light-mediated CarH dissociation (photolysis of the cofactor) it would be expected that the system will remain in the on-state as long as TtCBD-TetR-VP16 is present in the cell and that the system will only return to background levels once this protein has been degraded. Hence, the stability of TtCBD-TetR-VP16 seems to be the limiting factor in the reversibility of the system. In order to better quantify this process it is requested to experimentally determine the degradation rate of this protein.

Thank you for these comments. We added the label “only light groups (ON)” above the green line in Figure 3c to clarify that only samples from the light group (ON) were exposed to green light. We also clearly indicated groups that were exposed to green light (ON) and were kept in dark (OFF) in figure legend 3c. We also plotted each group separately in Supplementary figure 6a in order to clarify the setup.

In Figure 3c, the purple (OFF-ON) and black (OFF-OFF) groups were both kept in the dark for 24 h first (OFF). Hence, the level of leakiness after 24 h is almost the same (they overlap in a single line). After 24h, only the purple group (but not the black group) was exposed to light (ON) and thus levels of purple were induced accordingly (while levels of the black (OFF-OFF) group show the leakiness of the system). In a similar but opposite manner, blue and red were exposed to green light (ON) for 12 h. Therefore, SEAP levels at 12 h and 24 h are almost the same. After that, cells in the blue group were again exposed to light (ON-ON) leading to the highest SEAP levels of all groups at 48 h, whereas the red group was transferred to the dark (ON-OFF) resulting in significantly lower SEAP levels at 48 h. We have also performed an

additional experiment in a continuous culture without reseeding or changing media to further investigate the reversibility of the system (Supplementary Figure 6b).

Regarding the transactivator (TtCBD-TetR-VP16) degradation rate, we studied the protein half-life of TtCBD-TA by western blotting and described the results in page 4 and Supplementary figure 6c and 6d. Furthermore, we also show that the half-life of the transactivator can be regulated by fusing a degron domain to the TtCBD-TA (please see Supplementary figure 6c and 6d). Further details are provided in the methods section (page 10). In addition, from a safety point of view, it's important to note that Glow Control can be turned off at any time simply by addition of the clinically licensed antibiotic doxycycline, which targets the TetR-domain and releases TtCBD-TA from P_{TET} (see text and Fig. 3d).

4. How long do cells survive/remain functional in the microcapsules? Do they lose activity over time and would there be the need to transplant them regularly? How could this be done practically? Repeated surgery or other less invasive measures? Please provide a perspective for a real-world application in the discussion.

We thank Reviewer 1 for this feedback. In this paper, we assessed the functionality of the transplanted cells *in vivo* over 12 days (figure 6 c and e). Furthermore, we confirmed the viability and functionality of implanted microencapsulated cells that were retrieved from mice after 12 days *in vitro* (Supplementary Figure 15). More recently, we demonstrated that alginate-microencapsulated HEK293T cells subcutaneously implanted in mice could survive and remain functional for 21 days (Bai et al., Nature Medicine, 2019). Although alginate-based microencapsulation has been successfully used for transplantation of human islets in clinical trials (Jacobs-Tulleneers-Thevissen et al, Diabetologia, 2013), biomaterials researchers are actively working on novel materials and strategies to improve implant efficiency and viability of cells upon transplantation. For example, enhanced capsule materials with a lower susceptibility to pericapsular fibrotic overgrowth (PFO) have been shown to increase the functionality of microencapsulation-based cell implants *in vivo* (Vegas et al., Nature Medicine, 2016). Additionally, cell-based therapy can be prolonged by periodic injection of cells over time (Ozeki N et al, Osteoarthritis Cartilage., 2016) or by *in situ* refilling of the implant with fresh cells within the body (Krawczyk et al., Science 2020). Following your advice, we also stated in our discussion section (page 7) that long term follow-up studies with novel strategies and materials will be needed for future implementations.

Minor points

1. Figure 2a shows the system with C-terminally tagged TtCBD (pMMZ284) which was also used for further experiments. Anyhow in Figure S5 it is shown that the N-terminally tagged version (pMMZ410) works better. Why was the C-terminally tagged version used then?

Thank you for spotting this! We had mislabeled the bar graph in Supplementary Figure 5, and have now corrected it.

2. Figure 2c the "N" for the native GFP is not readable. All in all, a bigger font in the Figures would make reading more comfortable.

We agree, and have increased the font size as suggested.

3. Figure 3a and b: Why is the fold change at 24 h smaller than after 12 h and why is this shown in 2 different experiments? Wouldn't make one continuous experiment more sense?

In Figure 3a (old version) cells were exposed to 12 h light for two days in a row and their SEAP levels were measured after 48 h. However, in figure 3b, cells were illuminated for 12 h/day and their SEAP levels were measured every 24 h. To avoid confusion, we have replaced the old Fig. 3a with a new figure showing the in vitro kinetics of the Glow control system (Figure 3a new version). Here, cells were exposed to light and their SEAP levels were measured continuously for 48h.

4. Figure 3h is too small and therefore not readable.

We have increased the size (now Figure 4b).

5. Supplementary Figure 2b: It looks like the TtCBD-TetR-VPR-mCherry construct shuttles into the nucleus upon green light exposure. How is that to be explained? Is there an NLS sequence in the protein?

In Glow Control cells, the TtCBD-transactivator (TA) is bound to the plasma membrane in the dark. Green light induces dissociation of the transactivator from the plasma membrane and triggers its translocation to the nucleus. We did not add an NLS signal sequence because TetR inherently translocates to the nucleus. Indeed, commercial suppliers of the Tet system note that removing the NLS can reduce leakiness ([http://www.takara.co.kr/file/manual/pdf/PT3898-1\[5\].pdf](http://www.takara.co.kr/file/manual/pdf/PT3898-1[5].pdf)).

Reviewer #2 Remarks to Author:

In this work, the authors engineered mammalian cells to respond to green light and in-turn induce the expression of transgenes. The application of this green light-inducible switch was demonstrated in diabetic mice where human GLP-1 expression was triggered by the green light from wearable electronic devices. Overall, the manuscript is reasonably well-written and the data is well-presented.

We thank Reviewer 2 for the valuable comments and feedback on our manuscript. We have revised the manuscript accordingly, and our detailed responses are as follows.

Major comments

1. For SEAP quantification assays, was there any normalization done (e.g. cell number)? If so, please specify.

We followed a standard protocol for SEAP measurement and seeded the same number of cells per well in each experiment, so normalization was not done. The data are actual values without subtraction of SEAP background levels. We have addressed this point in the method section (page 9).

2. In figure 2b: Very low levels of SEAP were observed for MT-VP16+, MT- -VP16+, MT+- VP16- when the green light was pulsed during the assay. Please comment on the reasons which may have led to this observation.

We think tagging super charged GFP to TtCBD-transactivator (TtCBD-TA) may disturb the function of this complex, possibly due to steric hindrance of complex formation and/or hampered translocation of the TtCBD-TA to the nucleus due to the larger size or altered charge of the complex (Colwell et al., "Charge as a Selection Criterion for Translocation through the Nuclear Pore Complex", 2010).

3. In figure 2d: Please state which plasmid constructs were used for the various cell lines.

Thank you for the suggestion. We have added the name of each plasmid in figure legend 2d.

4. In figure 2: Based on the results, please provide a detailed explanation of the specific choice of pMMZ284 (PSV40-Myr-TtCBD-nGFP-pA) which was employed for subsequent experiments.

Based on a consideration of the results in figure 2b and Supplementary figure 5 (please note we have fixed a mislabeling in the bar graph of this figure), we concluded that pMMZ248 and TtCBD-TA (TtCBD-TetR-Vp16) provided the lowest level of leakiness and highest fold induction. Thus, we used this combination in subsequent experiments. We have noted this on page 4.

5. In figures 3c and 3d: The reversibility of the system is not clearly demonstrated. For instance, the reason for reseeded cells at 24h and a subsequent difference in response after the cells were reseeded, is not clear. Also, how is reversibility demonstrated in these figures based on the suitability of SEAP as a reporter? Please address these concerns with detailed explanations of the observed data.

Thank you for pointing out the need for additional explanation. Since SEAP is a stable protein, we reseeded the cells/exchanged media to completely re-set the experiment for the "ON" and "OFF" states. Our methodology, including a reseeded strategy, is in line with many reported reversibility experiments using SEAP (such as, but not limited to: Yin et al., Sci. Transl. Med. 2019; Liu et al., Cell, 2018; Saxena et al., PNAS, 2016, Ye et al., Science 2011). We have also confirmed the reversibility of the Glow Control system both in vitro and in vivo by measuring hGLP1 (therapeutic protein), as shown in Figure 6h and Supplementary figure 15d.

a. Please provide experimental data with statistical analysis showing reversibility of the system without reseeded of cells.

We have carried out a reversibility experiment in a continuous culture without reseeded cells or exchanging media. The results of this experiment and related statistical analysis are provided in Supplementary figure 6b.

b. Please provide experimental data with statistical analysis showing reversibility with doxycycline up to 48h instead of 42h.

Thank you for pointing this out. We have fixed the mislabeling in Figure 3d (48 h instead of 42 h). We have also included statistical analysis for Figure 3d.

6. In figure 3e: Please define and provide details on how this result demonstrates the targeted level of precise fine-tuning.

We have added an explanation that precise fine-tuning can be achieved by programming light-related parameters, e.g., using different exposure time and light intensity, on page 4.

7. In figure 3h: The design and the constructs used to achieve the data shown are not clear. Please provide a more in-depth analysis of the same in the text.

We have provided a new schematic figure (now in Figure 4b) which we believe is clearer and more informative. We have also provided more details about the experimental design on page 4.

8. In Figure 4d, active hGLP-1 levels were measured for Glow-controlhGLP-1 cells with the illumination schedule 545nm, 12h/day, 15s ON/45s OFF. This illumination schedule differs from the subsequent in vivo experiments where the illumination pattern was 30s ON/30s OFF. Please provide relevant in vitro experimental data for active hGLP-1 levels from Glow-controlhGLP-1 cells with illumination pattern of 30s ON/30s OFF.

We thank Reviewer 2 for pointing this out. We have repeated the in vitro experiment on Glow Control cells and stimulated them in the same setting used for the in vivo experiment (30s ON/30s OFF). A new version of the in vitro production of hGLP1 by Glow Control cells is provided in Figure 6b. We also revised the figure legend accordingly.

9. In figures 4e-4i: Statistical significance between the groups is not well indicated. Please provide statistical analysis on the results and indicate significance in these figures.

We thank Reviewer 2 for this feedback. We have indicated the statistical significance of differences between compared groups in figure 4e-4i (now figure 6c-h).

10. In figure 4e: Blood hGLP1 levels were measured at Day 1 after cell implantation before illumination (T0) and at 48 h (T48) after the start of illumination. To further understand the time course of the hGLP1 level, provide the hGLP1 levels at Day 1 and 3 after the start of illumination.

We thank Reviewer 2 for this suggestion. We have repeated this experiment and provided the in vivo kinetics of hGLP1 over 12 days in Figure 6c. The results include T0, T6h, T12h, T24h, T48h, T72h, T168h, T288h in T2D mice exposed to pulsed green light (12 h/day).

11. In vitro data demonstrating reversibility of Glow-controlhGLP-1 cells is lacking to support Figure 4j. Please provide relevant experimental data with statistical analysis.

We have done a new in vitro reversibility experiment under the same conditions as used in the in vivo experiment in Figure 4j (now Figure 6h). The results are provided in Supplementary figure 15d.

12. In order to provide a clear understanding of the translational capability of this technology, please discuss its advantages in the conclusion section. For instance, as the bioengineered cells need to be implanted under the skin:

- What is the half-life of the implanted cells? If it needs to be constantly re-implanted? If so, how often is it required?
- What are the safety issues involved in implanting these bioengineered cells under the skin?

We have provided experimental data on the viability and functionality of the microencapsulated cells recovered from mice after 12 days (Supplementary figure 15). We also added a discussion of issues around cell-based therapies, including necessary long-term studies, more compatible materials for cell encapsulation, and clinically relevant engineered cells other than HEK293T cells, on page 7.

- Upon switching on the light, how long does it take before a significant level of GLP1 can be detected in the blood stream?

Based on figure 6c, hGLP1 is detectable in the blood stream of mice after 12 h of green light illumination.

Minor comments

1. Please include all plasmids used and designed in the supplementary table 1. pMMZ429 was not found in the table.

Thank you for pointing this out. We have included pMMZ429 in the plasmid list.

2. There is a repeat pMMZ351 listed found in the supplementary table 1. Please revise this appropriately.

The repetition has been removed.

3. In figure 1c: It shows that SEAP level (Myr-TtCBD-, TtCBD-TetR-VPR+, Reporter+) under green light is higher (2 ×) than that in Dark. Please explain the reasons for the difference in SEAP level.

We thank Reviewer 2 for this comment. In Figure 1c, we showed that expression of myristoylation signal peptide-TtCBD (MT) is crucial to reduce the leakiness of the system and consequently to increase fold induction. In the absence of MT, aggregation of TtCBD-transactivators (TtCBD-TA) in cis-format (TtCBD-TA/TtCBD-TA) is believed to induce formation of complex structures in the cytoplasm, which are not readily translocated to the nucleus (most likely due to their large size) in the dark state. This complex can be monomerized upon illumination and the monomers induce gene expression (2x fold induction). We think that in the dark, there are still some free TtCBD-TA monomers that are not incorporated into such complexes, and which can still translocate to the nucleus without induction, resulting in leakiness for

the dark samples. Conversely, larger amounts of MT in Glow Control (TM⁺/Transactivator⁺/reporter⁺) would reduce the amount of free TtCBD-TA and therefore reduce the level of leakiness.

4. In References, appropriate and consistent abbreviations of journal names (e.g. Trends in Biotechnology, Nature Reviews Molecular Cell Biology) should be used.

Thank you. We have ensured that all the references are in the format of Nature Communications.

5. Throughout the manuscript, appropriate and consistent units such as days, hours, seconds (page 4, page 10) should be used.

Done as requested.

Reviewer #3 (Remarks to the Author)

The authors have constructed a novel system using TtCBD, where optogenetic stimulation of genetically engineered cells induces a transgene expression. They have validated the system composed of genetically engineered hGLP-1 producing cells implanted subcutaneously in an diabetic mouse model and a green-light-emitting device including Apple Watch, showing robust improvements in their metabolic parameters.

First of all, the results presented here seem to be another version of what was already reported by the authors (Science, 2011). While the system reported here might provide novelty in terms of handiness for clinical use as illustrated by the use of Apple Watch, the reviewer feels that it offers limited advantages over its precedent version as well as other currently available therapeutic tools such as islet transplantation or sensor-augmented put therapy (SAP).

We thank Reviewer 3 for these comments and feedback on the manuscript. We believe the major advantage of Glow Control over previously designed light-inducible cells is its ability to program therapeutic engineered cells non-invasively and remotely using a clinically-licensed green LED which already exists in commercially available smart watches. Progress in mobile health technology is extremely rapid (Sim, N Engl J Med, 2019), and we think technologies such as our Glow Controlled-cell based therapy represent an important step towards practical digital medicine (Dixon et al, Nat. Commun. 2021).

The potential limitations and other concerns are as following:

1. In the diabetic mouse model treated with the system, what is the kinetics of serum hGLP1 concentration over 12 days ?

We have examined the in vivo kinetics of hGLP1 in T2D mice over 12 days and provided the results in a new Figure 6c.

2. Related to 2, as deduced from Supplementary Fig. 14, the cell number is speculated to fluctuate over time due to replication or apoptosis. This fluctuation would not only limit the practical durability of this

system but also lead to instability of the transgene expression, leading to substantial problems. Could the authors overcome this issue?

We thank Reviewer 3 for raising this issue. The functionality of microencapsulated cells can be reduced over time, for example owing to fibrotic events that may occur after transplantation of the capsules. However, novel capsule materials with a lower susceptibility to pericapsular fibrotic overgrowth (PFO) can ameliorate this problem (Vegas et al., *Nature Medicine*, 2016). In addition, the functionality of cell-based therapy can be prolonged by periodic injection of cells into the implant over the course of the therapy (Ozeki N et al, *Osteoarthritis Cartilage.*, 2016) or alternatively by in situ refilling of fresh cells in special holder places within the body (Krawczyk K et al., *Science* 2020). We have pointed out these issues, all of which are areas of active research in the field, in our discussion section and also mentioned that long-term follow-up studies are needed for future implementations (page 7).

3. Given the wavelengths used in the experiments, it could be possible that the engineered cells might express the transgene without green-light stimulation when exposed to illumination or sunlight. The authors should discuss this issue.

We agree this is an important point. However, we kept un-illuminated mice in the mouse facility with periodic and standard day/night lights and have not observed transgene expression in any of the un-illuminated groups. Additionally, Glow Control cells would be implanted at a site that is normally covered by the smart watch, which would protect the Glow Control cells from direct ambient light. We have mentioned this point in the discussion (page 7).

4. The detailed description of hGLP1 is lacking. Is it a modified version so that it would not be cleaved by DPP-IV? In that case, is the ELISA kit able to detect the modified GLP-1?

We thank Reviewer 3 for raising this point. We used a modified version of human GLP1 (7-37) harboring two main changes which increase the half-life of the peptide; i) Ala in position 8 is substituted to Gly rendering the peptide resistant to DPP-IV (Parsons et al., *Gene Therapy* 2007; Burcelin et al., *Metabolism* 1999) and ii) GLP1 is fused to the Fc part of IgG (Ye et al, *Science* 2011; Xie, *Science* 2016). We clarified this in both the text (page 5) and plasmid list (page 22 in supplementary file). This modified version of GLP1 was quantified with the same ELISA kit that was used in other publications (e.g., Xie et al., *Science* 2016; Xue et al., *Mol. Ther.* 2017).

5. Related to 4, if the authors are using naive human GLP-1, the ELISA kit for GLP1 should not be specific for this transgene, as it detects naive GLP-1 derived from multiple species. In that case, the Y axis title of Fig 4d and 4e as well as the relevant texts should be amended.

Thank you. We have changed “active hGLP1” to “active GLP1” in Figures 6b, 6c and 6h, respectively.

6. In Fig4b and 4h, the comparison should be performed using One-way ANOVA with post hoc analyses.

We thank Reviewer 3 for this helpful feedback. We have re-analyzed figures 4b (now Figure 5d) and 4h (now Figure 6f) using one-way ANOVA and amended the method section as well as figure legends accordingly.

7. In Fig4f, 4g, 4i and 4j, the comparison should be performed using ideally Two-way ANOVA (or alternatively One-way ANOVA) with post hoc analyses.

We have re-analyzed these figures (now Figures 6d, 6e, 6g, and 6h) using two-way ANOVA and amended the method section as well as figure legends accordingly.

Reviewers' Comments:

Reviewer #1:

Remarks to the Author:

In the revised version the authors have adequately addressed my concerns. Especially the characterisation with regard to reversibility and half life of the transcription factor now provide important information on the functionality and performance of the system. The authors further discuss challenges ahead and provide a roadmap of how such wearables could become clinical reality.

Reviewer #2:

Remarks to the Author:

All the comments have been addressed sufficiently. Minor suggestions are:

1. Regarding safety issues involved in implanting bioengineered cells, while the points highlighted in the discussion such as cell encapsulation technologies are relevant, they do not directly address the safety and perception issue of actual implantation of bioengineered cells. It would be insightful if suggestions for future work that address those concerns are highlighted.
2. On Page 5, the text referencing Figure 5d states that SEAP levels were higher when programmed for 15s of illumination. This does not tally with the caption for Figure 5d which states that programmed LED case was programmed for 30s of illumination. Please rectify.
3. Some references in the discussion are inappropriately placed (e.g. after the full stop of sentences). Please ensure that references are appropriately cited.
4. 3rd line in the discussion "...cell-based therapies for disease management or lifestyle improvement." can be rephrased for clarity.

Reviewer #3:

Remarks to the Author:

All the points raised by the reviewer have been properly addressed either by the data or by discussion.

Revision of NCOMMS-20-43170-A – Point-By-Point Response

We are very grateful to all the referees for working with us on this manuscript, and our point-by-point responses are given below.

Reviewer #1 Remarks to Author:

In the revised version the authors have adequately addressed my concerns. Especially the characterization with regard to reversibility and half-life of the transcription factor now provide important information on the functionality and performance of the system. The authors further discuss challenges ahead and provide a roadmap of how such wearables could become clinical reality.

We are glad that we could address your concerns and appreciate your favorable opinion.

Reviewer #2 Remarks to Author:

All the comments have been addressed sufficiently. Minor suggestions are:

We thank the reviewer 2 for their positive evaluation of our revised manuscript and our detailed responses are as follows.

1. Regarding safety issues involved in implanting bioengineered cells, while the points highlighted in the discussion such as cell encapsulation technologies are relevant, they do not directly address the safety and perception issue of actual implantation of bioengineered cells. It would be insightful if suggestions for future work that address those concerns are highlighted.

We have addressed the immunogenicity issues upon implantation of bioengineered cells and also proposed possible solutions in our discussion (please see page 7).

2. On Page 5, the text referencing Figure 5d states that SEAP levels were higher when programmed for 15s of illumination. This does not tally with the caption for Figure 5d which states that programmed LED case was programmed for 30s of illumination. Please rectify.

Thank you for pointing this out. We fixed the caption for Figure5d.

3. Some references in the discussion are inappropriately placed (e.g. after the full stop of sentences). Please ensure that references are appropriately cited.

We have cited the references in appropriate places as requested.

4. 3rd line in the discussion "...cell-based therapies for disease management or lifestyle improvement." can be rephrased for clarity.

We have revised the corresponding part as suggested.

Reviewer #3 (Remarks to the Author)

All the points raised by the reviewer have been properly addressed either by the data or by discussion.

We appreciate the reviewer's favorable comment on our revised manuscript.